# Comprehensive Analysis of Seasonal and Geographical Variation in UVB Radiation Relevant for Vitamin D Production in Europe

**DOI:** 10.3390/nu14235189

**Published:** 2022-12-06

**Authors:** Tarinee Khanna, Rasha Shraim, Masa Zarkovic, Michiel van Weele, Jos van Geffen, Lina Zgaga

**Affiliations:** 1Department of Public Health and Primary Care, Institute of Population Health, Trinity College Dublin, D24 DH74 Dublin, Ireland; 2The SFI Centre for Research Training in Genomics Data Sciences, University of Galway, H91 TK33 Galway, Ireland; 3Institute of Social and Preventive Medicine, University of Bern, 3012 Bern, Switzerland; 4Royal Netherlands Meteorological Institute, 3731 GA De Bilt, The Netherlands

**Keywords:** ultraviolet B (UVB) radiation, Europe, vitamin D, public health, vitamin D supplementation, sun exposure

## Abstract

Dermal synthesis, following sun exposure, is the main source of vitamin D. This study characterizes ambient UVB radiation relevant for vitamin D production in Europe. A biological weighing function was applied to data from the Tropospheric Emissions Monitoring Internet Service (TEMIS) for 46 capital cities over an 18-year period (2004–2021) to isolate wavelengths relevant for vitamin D production (D-UVB). Cumulative and weighted D-UVB (CW-D-UVB) were calculated to approximate seasonal vitamin D accumulation and diminution. Monthly 25(OH)D concentration measurements were extracted from published reports. All data were analyzed by location and time. Despite a moderate latitudinal range (35–64° N), we observed large—up to five-fold—regional differences: the highest mean diurnal D-UVB dose of 5.57 kJ/m^2^ (SD = 3.55 kJ/m^2^) was observed in Nicosia (Cyprus) and the lowest in Reykjavik (Iceland, 1.16 ± 1.29 kJ/m^2^). Seasonal differences in diurnal D-UVB dose were even more pronounced, with a median 36-fold difference between annual peak and trough depending on a location (range: 10- to 525-fold). The mean duration of “vitamin D winter” was 126 days but varied widely (4 to 215 days). Monthly CW-D-UVB and 25(OH)D changes were very strongly correlated: the changes in 25(OH)D concentration increased by 12.6 nmol/L for every 100 kJ/m^2^ increment of CW-D-UVB in population-based studies (r^2^ = 0.79, *p*-value = 1.16 × 10^−37^). Understanding the differences in D-UVB radiation can help understand determinants of vitamin D status and guide region- and season-specific safe and effective sunlight exposure recommendations and vitamin D supplementation guidelines.

## 1. Introduction

A substantial body of evidence suggests that vitamin D is essential for maintaining health, and deficiency has been linked with numerous acute and chronic diseases [1]. However, vitamin D deficiency remains common worldwide. In Europe, 40% of the population were found to be deficient (25-hydroxyvitamin D [25(OH)D] < 50 nmol/L) and 13% severely deficient (25(OH)D < 30 nmol/L) [2].

Dermal synthesis following exposure to sunlight has long been recognized as the primary source of vitamin D for most people. In a photochemical reaction, the UVB component of solar radiation has the potential to convert 7-dehydrocholesterol (7-DHC) in the skin into pre-vitamin D_3_. Subsequently, pre-vitamin D_3_ isomerizes spontaneously into vitamin D_3_ [3]. It is estimated that as much as 90–100% of the daily requirements can be attained by UVB-induced vitamin D synthesis in the skin—even at European and North American latitudes [4,5]. Notwithstanding the fact that level of vitamin D synthesis is modified by numerous personal characteristics and behaviors that affect exposure and effectiveness of the downstream synthesis, the availability of ambient UVB is the root determinant of dermal vitamin D production [6,7,8,9]. This is best evidenced by the omnipresent seasonal co-fluctuation of vitamin D status and UVB in virtually every population, despite different personal characteristics, lifestyles, or built environment [2]. Similarly, numerous studies have consistently found an inverse association between vitamin D status and geographical location [10,11,12]: at higher latitudes, the UVB component of solar radiation is disproportionately more depleted when passing through the ozone layer. Consequently, the further an individual resides from the equator, the higher their risk of deficiency.

Understanding the ambient UVB dose—and thus the background dermal vitamin D production—at various locations and seasons, is key to formulating designer guidelines aimed at eradicating vitamin D deficiency, via oral vitamin D supplementation, food fortification or artificial light supplementation [13]. Moreover, as the broader benefits of sunshine for human health are being recognized, detailed understanding of solar radiation is critical for developing tailored sun exposure recommendations, with a view of maximizing benefit and minimizing harm. Unfortunately, despite being an easily accessible, key determinant of vitamin D status, precise measures of ambient UVB are rarely taken into account in epidemiological and clinical studies or considered in public health documents.

In this paper, we provide a detailed characterization of UVB dose relevant for vitamin D status in Europe. We describe regional and temporal trends, and provide data that can easily be adopted and used in future studies.

## 2. Methods

The list of 46 capital cities on the European continent was compiled. The latitude and longitude for each capital were established (Appendix A).

### 2.1. UVB Data and Derived Variables

#### 2.1.1. UVB Data

UV radiation data were extracted from the Tropospheric Emissions Monitoring Internet Service (TEMIS) database (www.temis.nl/uvradiation/UVdose.html (accessed on 17 October 2022); version 2.0) [14] for selected locations (2004–2021). This database has been used previously and is described in detail elsewhere [7,8]. Using satellite-based ozone measurements, UV radiation at the earth’s surface was computed every 5 min from sunrise to sunset and integrated over time to find the diurnal dose. Estimates were adjusted for numerous other variables, including surface elevation and surface UV reflectivity (UV albedo). Importantly, cloud cover attenuation was derived every 15 min from geostationary Meteosat Second Generation (MSG) satellite observations. Cloud cover was not available for some days (“blackout days”). In this case, the direct measurement was substituted with the mean diurnal UVB calculated for that day at that location, from available data (2004–2021). MSG observations were not available when the solar zenith angle (SZA) was more than 78° and therefore winter UVB dose could not be adjusted for cloud cover for some northern regions (Denmark, Norway, Estonia, Sweden, Latvia, Lithuania, Finland, Iceland, and Russia). In this case, mean cloud cover attenuation was approximated by a factor of 0.7 to adjust the UVB dose. Winter cloud data were missing from 10 days for Vilnius up to 105 days for Reykjavik around the winter solstice, when UVB dose is extremely low in those northerly regions and thus its impact on the UVB approximations is minimal. Data were available with a resolution of 0.25° longitude by 0.25° latitude. We only use the adjusted UVB provided by TEMIS as described above.

#### 2.1.2. Vitamin D-Specific UVB (D-UVB)

D-UVB dose was calculated by weighing UVB radiation dose at wavelengths that can induce cutaneous vitamin D production (290–315 nm) with the appropriate sensitivity function (peak synthesis occurs at 295–298 nm) [14,15,16].

#### 2.1.3. Vitamin D Winter

To calculate the duration of “vitamin D winter”, we assumed D-UVB dose of 1 kJ/m^2^ is the threshold below which vitamin D synthesis was negligible [11].

#### 2.1.4. Mean Diurnal Dose

Mean Diurnal Dose is calculated for each of the 365 days in a year (leap days were omitted). First, for any given day (e.g., 17th of June) available data were extracted from the TEMIS database, yielding 18 observations in most cases and mean D-UVB was calculated for that day, for each location of interest. Using these diurnal values, we calculated the *mean diurnal dose per month* and *annual mean diurnal D-UVB dose* for each location. Data averaged over the 18-year period is shown in this paper, except where we show temporal trends.

#### 2.1.5. Cumulative and Weighted D-UVB (CW-D-UVB) Dose

We calculated CW-D-UVB dose, to approximate the impact of seasonal and geographical differences on vitamin D level. More detail on the method can be found elsewhere [7,8,9]. Briefly, we extracted diurnal D-UVB dose over 135 days preceding the date of interest at a given location. To account for the fact that recent D-UVB exposure contributes more to current vitamin D level than past exposure (due to vitamin D metabolism), we weighted the diurnal D-UVB contributions before summing them up. This calculation provided estimates of CW-D-UVB for each day of the year for each location we studied.

### 2.2. Analysis

Descriptive statistics were used. Mean and standard deviation of D-UVB and CW-D-UVB were calculated for each location. The differences between locations as well as seasonal fluctuations were evaluated. We selected Athens and Dublin as southerly and moderate northerly locations for which we show more detail.

### 2.3. Association between CW-D-UVB and Monthly 25(OH)D

#### 25(OH)D and UVB

We searched the literature to find studies that report monthly 25(OH)D concentration. We found 15 studies [11,17,18,19,20,21,22,23,24,25,26,27,28,29,30] and extracted average monthly 25(OH)D concentration. We also calculated mean 25(OH)D for each month in the UK Biobank dataset (data not shown). Studies were classified as (A) healthy/population-based cohorts [11,17,20,21,23,24,25,26], (B) disease cohorts [18,19,22], or (C) laboratory samples/database [27,28,29,30], and analyzed separately. Based on the country the study was conducted in, we used average monthly CW-D-UVB data for the corresponding capital city from our file (except Faroe Islands where the mean between Oslo and Reykjavik was used). We calculated the difference in 25(OH)D and the difference in CW-D-UVB for each month relative to March (when 25(OH)D is typically the lowest), to somewhat account for the differences between the populations and focus on the change in 25(OH)D and CW-D-UVB instead. Finally, we assessed the relationship between the CW-D-UVB and the corresponding 25(OH)D concentration differences. 

## 3. Results

*Regional differences.* D-UVB dose over the year is shown in Figure 1A for selected cities (remainder are shown in Appendix A). We note large day-to-day variations in the diurnal dose (Figure 1B and Appendix A). Unsurprisingly, regions at lower latitudes were generally associated with higher levels of D-UVB radiation (Table 1). The highest mean annual diurnal D-UVB dose of 5.57 kJ/m^2^ (SD = 3.55 kJ/m^2^) was observed for Nicosia and the lowest for Reykjavik (1.16 kJ/m^2^ ± 1.29 kJ/m^2^). This equates to a cumulative annual D-UVB dose of 2033 kJ/m^2^ in Nicosia and nearly five times less, 423 kJ/m^2^, in Reykjavik. Variation in the duration of “vitamin D winter” was notable. This period lasted over 6 months in examined cities at latitudes above 59°, but only for about 2 months at latitudes below 40° (Figure 2). D-UVB dose also varied by longitude. Overall, radiation dose was higher in the east compared to the west. For example, Brussels (50.80° N, 4.36° E), Prague (50.08° N, 14.44° E), and Kiev (50.45° N, 30.52° E) are at a similar latitude, but radiation was highest in Kiev (mean diurnal dose: 2.72 ± 2.45 kJ/m^2^), the most eastern of the three, followed by Prague (2.64 ± 2.30 kJ/m^2^), then Brussels (2.43 ± 2.12 kJ/m^2^).

*Seasonal differences.* Notable seasonal differences were observed. The lowest levels of D-UVB radiation were observed in December and the highest in July, or June at some higher latitudes (Table 1, Figure 1A and Appendix A). Between annual peak and trough, a median 36-fold difference was observed (range: 10- to 525-fold). The relative difference between peak and trough month varied according to latitude: in Athens mean July dose was 13-fold that in December. However, the difference was more pronounced with increasing latitude: in Zagreb it was 30-fold higher, in London 50-fold higher, and in Oslo 205-fold higher.

### CW-D-UVB

The findings relating to CW-D-UVB, a measure approximating vitamin D status in the population, largely mirrored these regional and seasonal patterns. The most notable difference is that CW-D-UVB dose peaks/troughs about one-to-two months later and is smoother (Figure 3). Daily data over the 10-year period (2010–2020, Athens and Dublin) are shown in Figure 4.

We observed a very strong correlation between monthly CW-D-UVB difference and monthly 25(OH)D difference (reference month: March) with data from population-based studies or cohorts of healthy individuals. For every 100 kJ/m^2^ change in CW-D-UVB, change of 12.64 nmol/L in 25(OH)D concentration was observed (β-coeff = 0.126, r^2^ = 0.79, *p*-value = 1.16 × 10^−37^, Figure 5A). The correlation was weaker in cohorts of diseased individuals (β-coeff = 0.065, r^2^ = 0.33, *p*-value = 0.00024, Figure 5B) and in laboratory datasets (β-coeff = 0.042, r^2^ = 0.38, *p*-value = 9.23 × 10^−9^, Figure 5C).

## 4. Discussion

Despite the moderate latitudinal range in Europe (35–64° N), we observed large regional and seasonal differences in UVB radiation that can induce vitamin D synthesis (D-UVB) across the continent.

As expected, the highest mean daily D-UVB doses were recorded in June/July and the lowest in December/January [11,22,31]. Primarily due to its impact on SZA, latitude was found to be the principal determinant of D-UVB dose, with higher radiation observed at lower latitudes [11,32]. SZA is the angle between the sun and zenith, an imaginary vertical line drawn perpendicular to the ground [33]. It depends on the latitude, season, and time of the day [33]. At noon, in the summertime, and near the equator SZA is small, and D-UVB is most abundant because the UVB rays travel a lesser distance through the atmosphere and are less likely to get absorbed (in the ozone layer) or scattered by air molecules (Rayleigh scattering). The opposite is true for a large SZA: at dusk or dawn, in winter or at latitudes above 37°, D-UVB is depleted. The reason the differences are larger in summer than in winter is the fact that the D-UVB does not vary linearly with the SZA: the lower the SZA (summer vs. winter), the stronger the increase of D-UVB with decreasing SZA (north vs. south). Of note, eastern cities in Europe with similar latitudes tend to receive higher D-UVB dose compared to their western counterparts, but this is likely due to differences in weather (related to proximity to Atlantic Ocean) and altitude [34]. Thus, it is worth noting that while latitudinal effect might transfer to other regions, longitudinal effect would not.

Median duration of “vitamin D winter” in Europe was 126 days, but the differences were notable and varied from only 4 days in Nicosia and 38 days in Valletta to 187 days in Oslo and 215 days in Reykjavik. The **day-to-day** D-UVB dose varied more prominently in the summer when the absolute loss due to cloud cover is the greatest. For the same reason—there being more to lose—the annual oscillation in D-UVB dose was more prominent in regions with more sunlight. We did not observe any trends that would suggest UVB radiation was increasing or decreasing over the observation period, which is in agreement with other work [35]. 

### 4.1. Comparison of UVB Radiation Estimates to Previous Work

Our findings are in line with previous reports [11], despite utilizing different methods to estimate vitamin D effective UVB dose and different date range [36,37]. For example, yearly UVB was reported to be 4.8 ± 3.4 kJ/m^2^ in Athens and 2.1 ± 1.9 kJ/m^2^ in Ireland, while we found 4.78 ± 3.30 kJ/m^2^ and 2.08 ± 1.88 kJ/m^2^, respectively [11]. Likewise, the duration of “vitamin D winter” reported by the us and O’Neill et al. [11] are in agreement.

Countless studies have confirmed a clear relationship between UVB exposure and 25OHD increase [38]. Vitamin D has a reasonably long half-life in the body (1 to 2 months) [39], which explains why there is a 2-month lag between peak/trough UVB radiation and peak/trough vitamin D concentration [20]. Normally, vitamin D accumulates while UVB is abundant (summer), and reserves are used when dermal synthesis is poor (winter). To capture seasonal variability in 25(OH)D, we have previously developed a method of calculating cumulative and weighted D-UVB (CW-D-UVB) dose, that takes into account accumulation and diminution. We found CW-D-UVB more strongly correlated with vitamin D status than diurnal UVB at, or around, the time of vitamin D status assessment [7,8]. Interestingly, CW-D-UVB dose has been associated with health outcomes in previous studies [40,41,42].

CW-D-UVB alone explained 12.4% of the variance in 25(OH)D in a large sample. For context, 138 genetic loci associated with 25(OH)D status explained only 4.2% [40]. When we linked our CW-D-UVB data with monthly 25(OH)D reported in the literature by others, we found a very strong correlation. This is remarkable given that 25(OH)D data came from different populations residing in different countries, and no other adjustments were made. A notable increase in the monthly 25(OH)D difference of 12.64 nmol/L was found with an 100 kJ/m^2^ increase in the monthly CW-D-UVB, which is comparable to 10–20 nmol/L increase in 25(OH)D that we found in an older cohort from Ireland [8]. These findings highlight the value of CW-D-UVB measure for vitamin D studies. Given its strong impact on 25(OH)D, CW-D-UVB can aid in understanding the determinants of vitamin D status, or be used in prediction modelling with a view of developing personalized recommendations.

Of course, for any given level of ambient UVB radiation, factors such as exposed body surface area, sun-seeking behavior, skin pigmentation, and age, strongly impact dermal production and the increase in 25(OH)D concentration following exposure [8,43]. Moreover, the relationship between UVB and vitamin D production is not linear [43]. It is modified by factors such as the exhaustion of precursors (most notably 7-dehydrocholesterol), shift to production of other metabolites, and degradation [38]. Finally, factors unrelated to dermal synthesis such as vitamin D supplementation, BMI, diet, or baseline 25(OH)D level strongly impact vitamin D status.

Baseline 25(OH)D level was found to be a crucial determinant of the increase in 25(OH)D concentration after irradiation—the lower the baseline, the greater the increase [38]. Therefore, deficient individuals may benefit the most from natural sources, and conversely, the benefit (for vitamin D production) may be negligible in those with already high 25(OH)D concentration, since the benefit from extra solar exposure is modest at baseline levels above 60 nmol/L, and seems to disappear at levels above 160 nmol/L [44]. Increasingly, studies are uncovering additional benefits of moderate exposure to sunshine, and attention is turning to optimizing the pattern and duration of exposure with a view of maximizing benefit and minimizing harm. Here we described differences in the UVB dose according to the region and season, which should be taken into account when determining the right dose of natural sun exposure. Moreover, this work can inform guidelines concerning vitamin D supplementation, especially during the (local) vitamin D winter.

On a separate but very important note, understanding seasonal changes in UVB and vitamin D is also important for interpreting findings from vitamin D randomized controlled trials [45], because seasonal fluctuation can undermine the trial power and drive null-findings [46,47].

### 4.2. Strengths and Limitations

This is the first study to systematically characterize UVB radiation relevant for vitamin D production and vitamin D status across European continent, significantly extending the number of locations for which data are available. Findings are based on a notably long observation period and include 18 years of measured UVB data (2004–2021). D-UVB measure used in this work has been adjusted for a broad range of factors that affects UVB dose related to vitamin D synthesis. In terms of limitations, we included only one city for each country. The doses observed might not correlate closely with other regions, particularly for countries such as Croatia that spans multiple climates or Norway that covers a wide range of latitudes or stretches into extreme regions. Our data did not take into account the day-to-day variations in air pollution [14], which may affect the UVB dose, especially on extremely polluted days (haze or smog layer).

## 5. Conclusions

Very large seasonal and regional differences in D-UVB radiation relevant for vitamin D production were observed across Europe. Vitamin D in winter lasted for <2 months at low European latitudes and >7 months at high European latitudes. Monthly differences in cumulative and weighted D-UVB were very strongly correlated with monthly changes in mean 25(OH)D in different populations. The findings reported here have important implications for developing region- and season-appropriate sun exposure and vitamin D supplementation guidelines.

## Figures and Tables

**Figure 1 nutrients-14-05189-f001:**
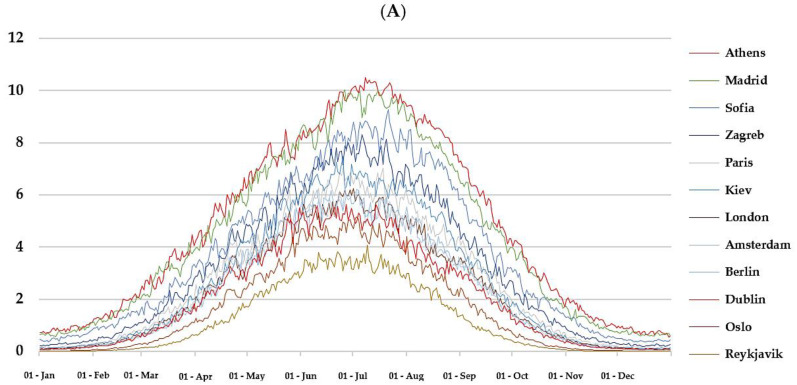
(**A**) Mean D-UVB dose for selected cities over the year. For any given day, data on UVB dose that can induce vitamin D synthesis (D-UVB) were extracted for the study period (2004–2021, thus yielding 18 observations in most cases) and mean D-UVB was calculated for that day of the year, for each location of interest. (**B**) Distribution of D-UVB dose by month for Dublin and Athens in 2020. **Footnote:** The data in panel B are plotted with the geom_density_ridges function (“ggridges” package), which estimate the density distribution from the provided D-UVB data and enables the visualization of changes in distributions over time. [citation: Claus O. Wilke (2021). ggridges: Ridgeline Plots in “ggplot2”. R package version 0.5.3. Available at https://CRAN.R-project.org/package=ggridges (accessed on 17 October 2022).

**Figure 2 nutrients-14-05189-f002:**
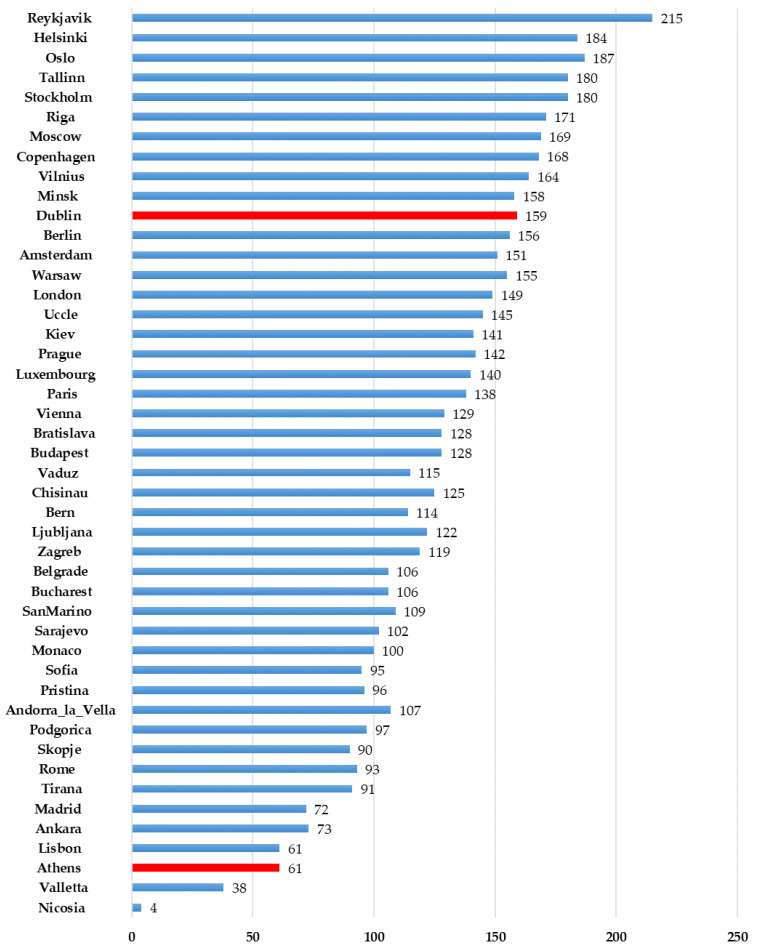
The duration of vitamin D winter in Europe. For any given day of the year, data on D-UVB dose were extracted for the study period (2004–2021, thus yielding 18 observations in most cases) and mean D-UVB was calculated for that day, for each location of interest. The number of days when average D-UVB dose is below 1 kJ/m^2^ is shown.

**Figure 3 nutrients-14-05189-f003:**
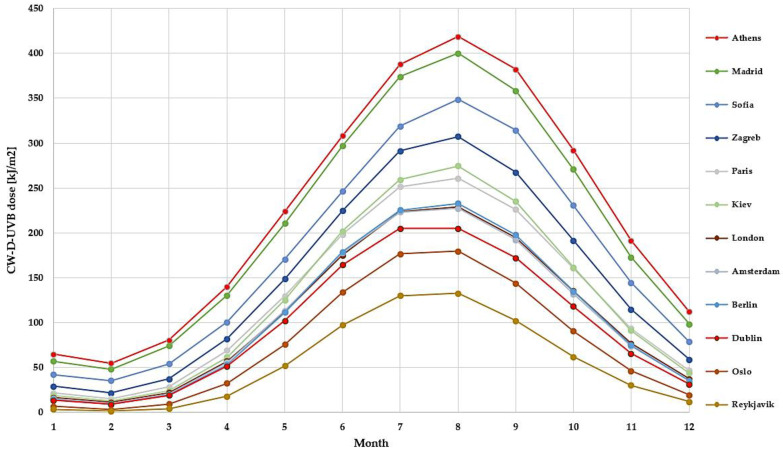
Mean monthly cumulative and weighted D-UVB (CW-D-UVB, a proxy of vitamin D status) for selected European capital cities.

**Figure 4 nutrients-14-05189-f004:**
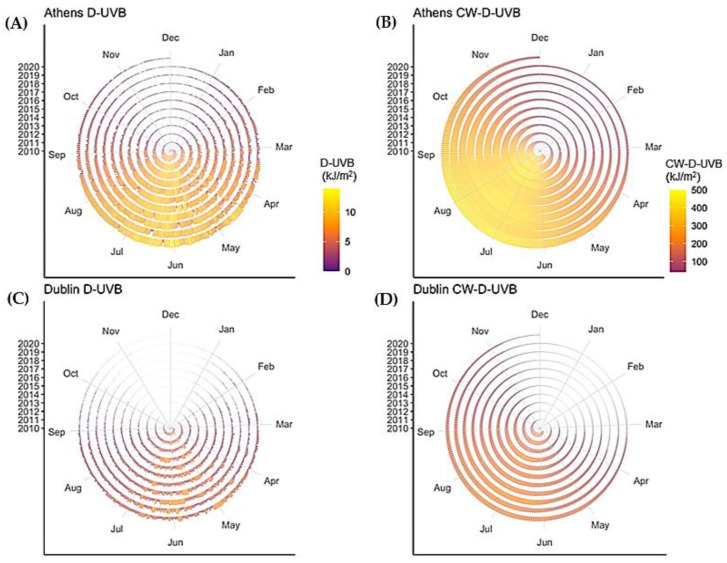
Observed diurnal D-UVB (**A**,**C**), calculated diurnal CW-D-UVB (**B**,**D**) and the highest CW-D-UVB reached for each year (including the date of the peak, (**E**)), in Dublin and Athens.

**Figure 5 nutrients-14-05189-f005:**
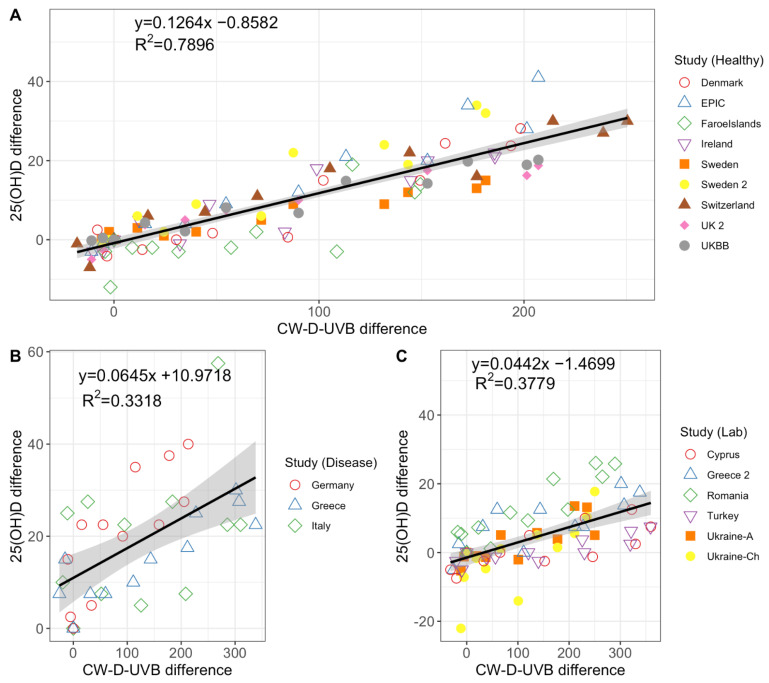
The relationship between the monthly difference in CW-D-UVB (versus March, the reference month) and the monthly difference in 25(OH)D (versus March, the reference month) for retrieved studies. CW-D-UVB data from this manuscript are used and linked with published reports examining 25(OH)D concentration in (**A**) healthy/population-based cohorts [11,17,20,21,23,24,25,26], (**B**) disease cohorts [18,19,22] or (**C**) laboratory samples/database [27,28,29,30]. Included studies are briefly described in Appendix A. **Footnote:** While Faroe Islands are Danish territory, due to the geographical location monthly CW-D-UVB differences were calculated using the Reykjavik and Oslo data (average between the two). One-third of the participants in the EPIC study had fractures, but the remainder are healthy controls, so study was included under population-based studies.

**Table 1 nutrients-14-05189-t001:** Mean diurnal D-UVB dose [kJ/m^2^] and standard deviation (SD) for European capital cities annually and by month (2004–2021; sorted by increasing latitude).

	Annual	January	February	March	April	May	June	July	August	September	October	November	December
Nicosia	5.57 (3.55)	1.23 (0.16)	2.23 (0.42)	4.07 (0.56)	6.24 (0.81)	8.31 (0.81)	10.33 (0.57)	11.06 (0.27)	9.51 (0.63)	6.78 (0.90)	3.84 (0.79)	1.92 (0.37)	1.13 (0.12)
Valletta	5.02 (3.38)	1.03 (0.11)	1.88 (0.42)	3.52 (0.72)	5.68 (0.69)	7.96 (0.54)	9.45 (0.69)	10.35 (0.34)	8.78 (0.71)	5.75 (0.89)	3.17 (0.66)	1.51 (0.32)	0.92 (0.08)
Athens	4.78 (3.30)	0.87 (0.12)	1.65 (0.36)	3.24 (0.60)	5.36 (0.72)	7.44 (0.51)	9.08 (0.64)	10.02 (0.28)	8.55 (0.58)	5.70 (0.95)	3.01 (0.74)	1.40 (0.36)	0.77 (0.11)
Lisbon	4.51 (3.06)	0.89 (0.14)	1.70 (0.34)	3.28 (0.53)	5.07 (0.71)	7.21 (0.55)	8.64 (0.53)	9.19 (0.26)	7.86 (0.64)	5.27 (0.75)	2.81 (0.75)	1.26 (0.27)	0.78 (0.06)
Ankara	4.66 (3.45)	0.71 (0.12)	1.39 (0.27)	2.79 (0.58)	4.99 (0.75)	7.21 (0.72)	9.05 (0.72)	10.42 (0.33)	8.76 (0.68)	5.68 (0.98)	2.83 (0.73)	1.23 (0.32)	0.62 (0.08)
Madrid	4.48 (3.20)	0.76 (0.14)	1.51 (0.33)	3.07 (0.55)	4.96 (0.65)	7.18 (0.56)	8.92 (0.67)	9.57 (0.28)	7.94 (0.67)	5.16 (0.83)	2.69 (0.75)	1.12 (0.27)	0.67 (0.06)
Tirana	4.01 (3.06)	0.57 (0.1)0	1.12 (0.27)	2.38 (0.61)	4.31 (0.67)	6.35 (0.71)	8.16 (0.75)	9.04 (0.44)	7.54 (0.66)	4.60 (0.87)	2.33 (0.51)	0.98 (0.29)	0.50 (0.05)
Rome	3.97 (3.01)	0.56 (0.09)	1.17 (0.27)	2.45 (0.61)	4.36 (0.54)	6.44 (0.68)	8.20 (0.70)	8.82 (0.38)	7.28 (0.68)	4.47 (0.76)	2.27 (0.52)	0.91 (0.24)	0.49 (0.04)
Skopje	3.84 (2.93)	0.57 (0.12)	1.22 (0.26)	2.24 (0.53)	4.12 (0.68)	6.07 (0.64)	7.81 (0.69)	8.61 (0.47)	7.27 (0.61)	4.41 (0.88)	2.21 (0.54)	0.90 (0.29)	0.44 (0.06)
Podgorica	3.90 (3.04)	0.50 (0.10)	1.01 (0.25)	2.26 (0.62)	4.31 (0.69)	6.31 (0.78)	8.14 (0.74)	8.79 (0.40)	7.35 (0.64)	4.45 (0.84)	2.19 (0.52)	0.86 (0.26)	0.43 (0.04)
AndorralaVella	3.69 (2.94)	0.41 (0.04)	0.84 (0.24)	1.98 (0.51)	3.70 (0.59)	5.90 (0.67)	7.73 (0.88)	8.57 (0.38)	7.04 (0.65)	4.45 (0.75)	2.31 (0.62)	0.77 (0.25)	0.37 (0.03)
Pristina	3.81 (2.94)	0.49 (0.12)	1.18 (0.26)	2.26 (0.57)	4.21 (0.72)	6.12 (0.66)	7.87 (0.59)	8.53 (0.44)	7.19 (0.65)	4.33 (0.91)	2.11 (0.54)	0.83 (0.27)	0.38 (0.06)
Sofia	3.75 (2.86)	0.55 (0.11)	1.16 (0.25)	2.24 (0.53)	4.11 (0.67)	5.96 (0.61)	7.55 (0.68)	8.40 (0.40)	7.15 (0.62)	4.33 (0.90)	2.09 (0.51)	0.84 (0.28)	0.42 (0.04)
Monaco	3.79 (2.93)	0.49 (0.09)	1.05 (0.28)	2.33 (0.52)	4.23 (0.66)	6.32 (0.65)	8.03 (0.61)	8.42 (0.32)	6.90 (0.62)	4.26 (0.71)	2.08 (0.58)	0.79 (0.21)	0.41 (0.04)
Sarajevo	3.41 (2.68)	0.43 (0.10)	0.99 (0.25)	2.07 (0.57)	3.75 (0.62)	5.47 (0.62)	7.21 (0.75)	7.71 (0.56)	6.47 (0.66)	3.82 (0.85)	1.85 (0.48)	0.70 (0.23)	0.34 (0.04)
SanMarino	3.55 (2.85)	0.41 (0.06)	0.88 (0.24)	2.08 (0.57)	3.93 (0.58)	5.9 (0.76)	7.68 (0.72)	8.19 (0.45)	6.56 (0.62)	3.96 (0.75)	1.82 (0.46)	0.67 (0.21)	0.35 (0.04)
Bucharest	3.54 (2.82)	0.42 (0.08)	0.96 (0.23)	2.04 (0.50)	4.00 (0.71)	5.95 (0.64)	7.47 (0.57)	7.98 (0.32)	6.71 (0.68)	4.03 (0.84)	1.80 (0.50)	0.64 (0.22)	0.32 (0.03)
Belgrade	3.39 (2.72)	0.34 (0.08)	0.90 (0.25)	1.97 (0.54)	3.86 (0.72)	5.66 (0.71)	7.28 (0.65)	7.70 (0.50)	6.31 (0.64)	3.77 (0.84)	1.76 (0.49)	0.64 (0.21)	0.27 (0.04)
Zagreb	3.19 (2.64)	0.31 (0.06)	0.72 (0.22)	1.80 (0.50)	3.66 (0.65)	5.41 (0.75)	7.14 (0.57)	7.35 (0.51)	5.94 (0.65)	3.48 (0.67)	1.58 (0.46)	0.51 (0.16)	0.24 (0.03)
Ljubljana	3.11 (2.61)	0.29 (0.06)	0.69 (0.23)	1.74 (0.45)	3.53 (0.59)	5.24 (0.76)	6.99 (0.64)	7.30 (0.47)	5.81 (0.67)	3.42 (0.67)	1.48 (0.42)	0.47 (0.14)	0.23 (0.02)
Bern	3.11 (2.49)	0.34 (0.08)	0.83 (0.27)	1.94 (0.48)	3.64 (0.62)	5.22 (0.85)	6.87 (0.70)	6.97 (0.49)	5.49 (0.54)	3.53 (0.68)	1.54 (0.44)	0.51 (0.16)	0.27 (0.03)
Chisinau	3.16 (2.66)	0.30 (0.06)	0.72 (0.20)	1.69 (0.46)	3.48 (0.72)	5.47 (0.69)	7.06 (0.52)	7.30 (0.32)	6.05 (0.66)	3.49 (0.74)	1.48 (0.49)	0.45 (0.18)	0.21 (0.02)
Vaduz	3.04 (2.45)	0.30 (0.07)	0.77 (0.25)	1.81 (0.49)	3.59 (0.62)	5.13 (0.76)	6.76 (0.66)	6.76 (0.51)	5.41 (0.60)	3.47 (0.68)	1.55 (0.42)	0.51 (0.16)	0.25 (0.02)
Budapest	3.09 (2.59)	0.25 (0.06)	0.64 (0.20)	1.69 (0.47)	3.63 (0.68)	5.34 (0.68)	6.92 (0.49)	7.09 (0.49)	5.78 (0.59)	3.39 (0.68)	1.47 (0.46)	0.47 (0.19)	0.19 (0.03)
Bratislava	2.97 (2.53)	0.22 (0.06)	0.64 (0.22)	1.60 (0.44)	3.57 (0.70)	5.20 (0.67)	6.81 (0.41)	6.90 (0.43)	5.42 (0.63)	3.26 (0.64)	1.32 (0.44)	0.40 (0.15)	0.17 (0.02)
Vienna	2.91 (2.45)	0.25 (0.06)	0.67 (0.21)	1.58 (0.44)	3.53 (0.71)	5.04 (0.65)	6.59 (0.47)	6.72 (0.40)	5.30 (0.61)	3.18 (0.64)	1.29 (0.44)	0.39 (0.14)	0.17 (0.02)
Paris	2.70 (2.29)	0.20 (0.05)	0.53 (0.17)	1.47 (0.40)	3.26 (0.59)	4.83 (0.66)	6.21 (0.45)	6.21 (0.39)	4.80 (0.56)	3.06 (0.65)	1.20 (0.37)	0.35 (0.12)	0.16 (0.02)
Luxembourg	2.63 (2.28)	0.17 (0.04)	0.48 (0.17)	1.39 (0.41)	3.18 (0.62)	4.79 (0.68)	6.18 (0.43)	6.11 (0.42)	4.64 (0.55)	2.94 (0.64)	1.09 (0.33)	0.30 (0.10)	0.13 (0.01)
Prague	2.64 (2.30)	0.19 (0.06)	0.53 (0.18)	1.36 (0.41)	3.18 (0.70)	4.76 (0.68)	6.23 (0.38)	6.16 (0.39)	4.79 (0.59)	2.86 (0.61)	1.08 (0.36)	0.29 (0.10)	0.15 (0.02)
Kiev	2.72 (2.45)	0.18 (0.05)	0.47 (0.16)	1.29 (0.43)	3.02 (0.67)	4.94 (0.75)	6.53 (0.44)	6.47 (0.28)	5.24 (0.69)	2.85 (0.67)	1.08 (0.41)	0.27 (0.11)	0.13 (0.02)
Brussels	2.43 (2.12)	0.15 (0.04)	0.42 (0.14)	1.26 (0.39)	2.95 (0.60)	4.54 (0.62)	5.75 (0.43)	5.63 (0.40)	4.24 (0.54)	2.69 (0.58)	1.00 (0.31)	0.27 (0.10)	0.12 (0.02)
London	2.32 (2.05)	0.14 (0.04)	0.40 (0.14)	1.19 (0.33)	2.77 (0.54)	4.31 (0.54)	5.56 (0.38)	5.49 (0.39)	4.06 (0.60)	2.54 (0.59)	0.91 (0.29)	0.26 (0.09)	0.11 (0.02)
Warsaw	2.43 (2.21)	0.14 (0.04)	0.40 (0.13)	1.14 (0.39)	2.82 (0.61)	4.46 (0.67)	5.97 (0.35)	5.81 (0.34)	4.55 (0.60)	2.55 (0.57)	0.91 (0.33)	0.22 (0.09)	0.10 (0.01)
Amsterdam	2.29 (2.06)	0.12 (0.03)	0.36 (0.13)	1.16 (0.39)	2.79 (0.59)	4.41 (0.55)	5.56 (0.36)	5.37 (0.45)	4.05 (0.59)	2.43 (0.57)	0.86 (0.30)	0.22 (0.08)	0.09 (0.01)
Berlin	2.32 (2.11)	0.11 (0.04)	0.38 (0.15)	1.09 (0.35)	2.77 (0.67)	4.36 (0.67)	5.68 (0.29)	5.48 (0.35)	4.25 (0.58)	2.44 (0.59)	0.84 (0.30)	0.21 (0.08)	0.08 (0.01)
Dublin	2.08 (1.88)	0.11 (0.03)	0.34 (0.13)	1.08 (0.35)	2.51 (0.48)	4.05 (0.65)	5.14 (0.28)	4.89 (0.42)	3.51 (0.44)	2.18 (0.51)	0.75 (0.27)	0.21 (0.08)	0.08 (0.01)
Minsk	2.28 (2.12)	0.10 (0.03)	0.36 (0.14)	1.14 (0.39)	2.59 (0.52)	4.28 (0.71)	5.68 (0.41)	5.54 (0.27)	4.30 (0.67)	2.24 (0.60)	0.72 (0.28)	0.16 (0.08)	0.07 (0.01)
Vilnius	2.16 (2.03)	0.09 (0.03)	0.31 (0.12)	1.04 (0.36)	2.46 (0.56)	4.12 (0.69)	5.44 (0.28)	5.27 (0.26)	4.04 (0.65)	2.12 (0.58)	0.66 (0.26)	0.14 (0.07)	0.06 (0.01)
Copenhagen	2.05 (1.97)	0.07 (0.02)	0.24 (0.10)	0.89 (0.33)	2.42 (0.57)	4.05 (0.57)	5.26 (0.22)	5.07 (0.39)	3.71 (0.63)	2.02 (0.57)	0.63 (0.25)	0.13 (0.05)	0.05 (0.00)
Moscow	2.10 (2.04)	0.08 (0.03)	0.29 (0.11)	0.91 (0.33)	2.25 (0.51)	4.13 (0.58)	5.40 (0.42)	5.28 (0.28)	4.08 (0.66)	1.90 (0.49)	0.58 (0.26)	0.12 (0.05)	0.06 (0.00)
Riga	1.98 (1.93)	0.06 (0.02)	0.25 (0.11)	0.90 (0.33)	2.21 (0.56)	3.86 (0.59)	5.12 (0.28)	5.04 (0.34)	3.66 (0.65)	1.82 (0.52)	0.53 (0.24)	0.10 (0.05)	0.04 (0.00)
Stockholm	1.76 (1.8)	0.03 (0.01)	0.15 (0.07)	0.66 (0.28)	1.93 (0.51)	3.49 (0.53)	4.77 (0.28)	4.69 (0.34)	3.20 (0.58)	1.54 (0.46)	0.40 (0.19)	0.07 (0.03)	0.02 (0.00)
Tallinn	1.76 (1.82)	0.04 (0.01)	0.15 (0.07)	0.66 (0.27)	1.90 (0.50)	3.59 (0.59)	4.79 (0.25)	4.71 (0.40)	3.26 (0.66)	1.48 (0.46)	0.39 (0.19)	0.07 (0.03)	0.02 (0.00)
Oslo	1.65 (1.7)	0.03 (0.01)	0.15 (0.07)	0.64 (0.27)	1.81 (0.46)	3.26 (0.55)	4.57 (0.25)	4.43 (0.35)	2.97 (0.53)	1.42 (0.49)	0.36 (0.17)	0.06 (0.03)	0.02 (0.00)
Helsinki	1.75 (1.84)	0.03 (0.01)	0.14 (0.07)	0.61 (0.27)	1.81 (0.48)	3.55 (0.57)	4.79 (0.30)	4.81 (0.40)	3.26 (0.71)	1.44 (0.46)	0.36 (0.18)	0.06 (0.03)	0.02 (0.00)
Reykjavik	1.16 (1.29)	0.01 (0.01)	0.06 (0.03)	0.32 (0.15)	1.15 (0.34)	2.46 (0.36)	3.41 (0.23)	3.28 (0.29)	2.15 (0.45)	0.82 (0.28)	0.20 (0.10)	0.03 (0.01)	0.01 (0.00)

First, for any given day, data on D-UVB dose were extracted for the study period (2004–2021, thus yielding 18 observations in most cases) and mean D-UVB was calculated for that day, for each location of interest. This reduced the number of datapoints to 365. Next, monthly and annual mean and SD were calculated using this data.

## Data Availability

All data used in this paper are publicly available (https://www.temis.nl/) (accessed on 17 October 2022). We are happy to provide curated dataset upon request.

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
