# Peer review of "Comprehensive Analysis of Seasonal and Geographical Variation in UVB Radiation Relevant for Vitamin D Production in Europe"

_nutrients, 2022, doi:10.3390/nu14235189_

Round 1

Reviewer 1 Report

While the manuscript is well-written and clear, the figures have some room for improvement:

Fig. 1 a: seems to give the same information as Figure S3 for different cities. Styles could be matched.

Fig. 1 b: please add a unit of measurement to x-axis.

Table 1: is the number in brakes the respective standard deviation? please specify in the description of the table. 

Figure 3: different shades of violet/pink colors make it impossible to differentiate individual graphs.

Figure 4: Please add units of measurement.

Fig. 1 a + Figure S3: While the x-axis is clear, the authors could provide dates instead of the day numbers of the year at the x-axis for easier interpretation (as in Figure S4).

Figure S1: The figure could be removed, as not referred to in the text, and adding no substantial information to Table S1

Figure S2: The figure is quite overloaded, with different shades of violet colors making it hard to grasp the details

Figure S3: it is not clear on what premise the figure was divided into 4 sub-figures; whereas Figure S2 captures all cities.

Reviewer 2 Report

The manuscript explores the vitamin D effective radiation available across Europe using satellite ozone data and a radiative transfer model. This is then compared with 25(OH)D data extracted from literature. It illustrates the seasonal and latitudinal differences in D-UVB, and how this impacts on 25(OH)D. There is nothing particularly novel about this concept, but the changing D-UVB is well illustrated. The paper is suitable for Nutrients, but some revisions are necessary before publication. The text is not always suitably precise.

Abstract

No vitamin D data was captured, please rephrase.

Clarify exactly what changed regionally and seasonally. The 5-fold regional change would appear to be the annual average daily dose. This should be stated. However, the seasonal change in this same measure varies enormously, from about 10 to over 200. How was the value 20 achieved?

The sentence beginning ‘The highest diurnal D-UVB dose…’ is also confusing. From Table 1 it seems this refers to the mean (+/- SD) of the diurnal D-UVB across all days in the years 2004-2021, though it could also be the mean (+/-SD) of the annual mean diurnal D-UVB, where the SD refers not to all individual days but to the distribution of the means for each of the 18 years. Whichever it is, what is written is not suitably descriptive.

The 25(OH)D did not change by 9.6 nmol/L – no information is given about absolute values of 25(OH)D. The change in 25(OH)D increased by 9.6 nmol/L for each increment of cw-D-UVB.

Methods

Section 2.1 – write in the past tense.

Clarify what was taken directly from TEMIS, and what was adjusted by the authors. From what is written it would seem the 5 minute UVB data came from TEMIS, derived from satellite ozone data. Thereafter it is not clear whether the cloud correction was applied by TEMIS, or by the authors (‘we substituted the direct measurement’), and whether this was done with the temporal resolution of the MSG data, or applied only to the diurnal total. Also please indicate how many blackout days occurred i.e. what proportion of the data had to be substituted. What about aerosol?

Section 2.2

State and reference the action spectrum used for the weighting of D-UVB.

Section 2.3

These are not measurements. Please replace with data, or calculations.

2.3.1  …yielding 18 observations in most cases and average D-UVB was calculated for that day, for each location of interest.

With this addition, and the rest of the paragraph, the confusion in the Abstract is clarified. Note the abstract should be standalone and so still requires the adjustments already noted.

Section 2.4

The search was not very comprehensive e.g. Webb, A.R. et al (2010). The role of sunlight exposure in determining the vitamin D status of the UK white Caucasian adult population. Br J Dermatol.163, 1050-1055.

It should also be noted that most of the studies used were cross-sectional, not longitudinal, and were limited to the elder female (2), HIV patients (1), urologic patients (1). Only 2 used healthy working age adults from the general population. See later comments. It might be helpful to widen the search and try to find more data from healthy cohorts of the population.

Section 2.5  Southerly (sp)

Results

‘Average’ is used a lot in the text, figure captions etc. Please be more precise. I assume the values are the mean (not median) so replace ‘average’ with ‘mean’ throughout.

Section 3.1

Be clear, not ‘Average D-UVB dose’ but ‘Mean daily D-UVB dose’

What is the reason for the longitudinal differences? At a guess it is cloud cover with the cities at increasing distance from the ocean and source of moisture. You should have the data from the modelling to identify this. The longitudinal effect may not transfer to other regions of the world (while latitudinal effect will), so it should be mentioned.

Figure 1 caption – clarify Mean daily D-UVB, and (B) Distribution of daily D-UVB. Is this also over all days of the month in all years?

Figure 2 – is this the average number of days (all sky) across 18 years, or is it the data from clear sky UVB retrievals? The latter should be pretty consistent, but the former can vary with year to year changes in cloud cover.

Table 1 – clarify caption

Section 3.3

Be consistent, CW or cw or Cw? Check and correct for chosen version throughout manuscript.

There are no absolute values of 25(OH)D provided (see comment for abstract)

Most figures require an increased font size for axis labels and other labels within the figure. In particular Figs 1B, 3, 4 (especially 4E).

Figure 5 – difficult to distinguish the coloured dots, especially the dark green / blue.

A different line would result if the yellow and light green data were used. It is not possible to know which Swedish study is light green, but if these are the two studies with healthy, working age cohorts then that should be noted, since age, infirmity, illness and the medications taken for it can all impact on sun exposure and the vitamin D pathway. Please add the appropriate reference to each study location. See earlier comments about change in 25(OH)D, and also make this correction in section 4.

Please add the data from Webb et al (2010) and reanalyse outcomes as appropriate, bearing in mind the comments above.

Section 4

See earlier comments on 20-fold, and change in 25(OH)D.

Highest mean daily D-UVB dose (see previous comments)
